# Ceruloplasmin, Catalase and Creatinine Concentrations Are Independently Associated with All-Cause Mortality in Patients with Advanced Heart Failure

**DOI:** 10.3390/biomedicines12030662

**Published:** 2024-03-15

**Authors:** Wiktoria Smyła-Gruca, Wioletta Szczurek-Wasilewicz, Michał Skrzypek, Andrzej Karmański, Ewa Romuk, Michał Jurkiewicz, Mariusz Gąsior, Bożena Szyguła-Jurkiewicz

**Affiliations:** 1Student’s Scientific Society, 3rd Department of Cardiology, Faculty of Medical Sciences in Zabrze, Medical University of Silesia, 40-055 Katowice, Poland; wiktoriasmyla@gmail.com (W.S.-G.); jurkiewicz.m@yahoo.com (M.J.); 2Silesian Center for Heart Diseases, 41-800 Zabrze, Poland; 3Department of Biostatistics, Faculty of Public Health in Bytom, Medical University of Silesia, 40-055 Katowice, Poland; mskrzypek@sum.edu.pl; 4Department of Descriptive and Topographic Anatomy, Faculty of Medical Sciences, Medical University of Silesia, 40-055 Katowice, Poland; akarmanski@sum.edu.pl; 5Department of Biochemistry, Faculty of Medical Sciences, Medical University of Silesia, 40-055 Katowice, Poland; eromuk@gmail.com; 63rd Department of Cardiology, Faculty of Medical Sciences in Zabrze, Medical University of Silesia, 40-055 Katowice, Poland; mgasior@o2.pl (M.G.); bjurkiewicz@sum.edu.pl (B.S.-J.)

**Keywords:** oxidative stress, advanced heart failure, markers, coronary sinus blood

## Abstract

The role of oxidative/antioxidative system imbalances in advanced heart failure (HF) has not been fully investigated. The aim of this study was to identify factors associated with one-year mortality in patients with advanced HF, with particular emphasis on oxidative/antioxidative balance parameters. We analyzed 85 heart transplant candidates who were hospitalized at our institution for right heart catheterization. Ten milliliters of coronary sinus blood was collected to measure oxidative/antioxidative markers. The median age was 58 (50–62) years, and 90.6% of them were male. The one-year mortality rate was 40%. Multivariable logistic regression analysis revealed that ceruloplasmin (OR = 1.342 [1.019–1.770], *p* = 0.0363; per unit decrease), catalase (OR = 1.053 [1.014–1.093], *p* = 0.0076; per unit decrease), and creatinine (OR = 1.071 [1.002–1.144], *p* = 0.0422; per unit increase) were independently associated with one-year mortality. Ceruloplasmin, catalase, and creatinine had areas under the curve of 0.9296 [0.8738–0.9855], 0.9666 [0.9360–0.9971], and 0.7682 [0.6607–0.8756], respectively. Lower ceruloplasmin and catalase in the coronary sinus, as well as higher creatinine in peripheral blood, are independently associated with one-year mortality in patients with advanced HF. Catalase and ceruloplasmin have excellent prognostic power, and creatinine has acceptable prognostic power, allowing the distinction of one-year survivors from nonsurvivors.

## 1. Introduction

Oxidative stress is caused by an imbalance between the formation of reactive oxygen species (ROS) and the ability of biological defenses to remove or neutralize these reactive products [1]. With long-term exposure of cells and tissues to ROS, antioxidant systems are gradually depleted, which contributes to progressive cell and tissue damage via the formation of free radicals [1,2]. ROS causes negative changes in subcellular organelles and enzyme activity and reprogramming of gene expression, cell signaling, and metabolism [2,3]. Other harmful effects of free radicals are mainly related to damage to lipids, proteins, and genetic material, which constitute the basic building blocks of the cell. The consequences of ROS on cells are changes in the integration, conformation, folding, and aggregation of the cell membrane, which ultimately, despite many adaptive mechanisms, disrupt the normal functioning of the cells [3,4,5].

Notably, under physiological conditions, ROS act as signaling molecules to regulate many processes in the cardiovascular system and contribute to cardiovascular homeostasis through complex processes [1]. However, excessive and long-lasting increases in ROS generation lead to the exhaustion of antioxidant cell defense mechanisms and play an important role in the initiation and progression of endothelial dysfunction, premature apoptosis, interstitial fibrosis, cardiomyocyte hypertrophy, and extracellular matrix remodeling [5,6], which consequently contributes to the development and progression of cardiovascular diseases [2,6,7]. Advanced heart failure (HF) is associated with ischemia–reperfusion injury, increased neurohumoral activity, cytokine stimulation, and inflammation [6,7]. These factors are powerful stimuli that induce the production of free radicals in the heart and may cause damage to basic myocardial structures and functions, e.g., extracellular remodeling, the activation of hypertrophy signaling kinases and transcription factors, apoptosis, and calcium metabolism [6,7,8].

Considering the associations between oxidative stress and the induction of endothelial dysfunction, damage to cardiomyocytes, premature apoptosis, interstitial fibrosis, and extracellular matrix remodeling, we speculated that oxidative/antioxidative balance disorders lead to a cascade of changes that could cause the development and progression of HF. Therefore, in this study, we aimed to determine the associations between oxidative/antioxidative balance markers and all-cause mortality in heart transplant (HT) candidates. Furthermore, we analyzed the associations between routine laboratory parameters and patient outcomes in our study group.

## 2. Materials and Methods

### 2.1. Study Population and Data Collection

This was a prospective review of 154 patients with advanced HF underwent qualification for HT, who were hospitalized at our institution for right heart catheterization (RHC) between 2016 and 2017. All patients underwent a preliminary assessment according to the exclusion and inclusion criteria. A flow chart of the study design for the inclusion and exclusion criteria is presented in Figure 1.

The patients were treated optimally using maximum tolerated doses of drugs according to the current recommendations for HF management. Follow-up information was collected from the patients or their families during ambulatory care visits or telephone interviews. No patient was lost to follow-up. The endpoint of the study was defined as all-cause mortality during a one-year follow-up. The study protocol was approved by the ethics committee of the Medical University of Silesia in Katowice (specific ethics code KNW/0022/KB1/88/15). The study complied with the principles described in the Declaration of Helsinki. All patients provided informed written consent to participate in the study.

At the time of enrollment in the study, routine laboratory tests of peripheral blood samples, echocardiography, ergospirometric exercise tests, spirometry, and RHC were performed on all included patients.

RHC was performed using a Swan–Ganz catheter via the right internal jugular vein. During RHC, 10 mL of blood from the coronary sinus was collected from all patients. The protocol for blood collection from the coronary sinus was described in the study by Szczurek-Wasilewicz et al. [9].

### 2.2. Laboratory Measurements of Peripheral Blood

Determination of the complete blood count and hematological parameters was performed using automated blood cell counters (Sysmex XS1000i and XE2100; Sysmex Corporation, Kobe, Japan). Biochemical parameters were analyzed with a COBAS Integra 800 analyzer (Roche Instrument Center AG, Rotkreuz, Switzerland). The plasma concentration of the N-terminal prohormone of brain natriuretic peptide was measured with a commercially available kit from Roche Diagnostics (Mannheim, Germany) on an Elecsys 2010 analyzer with an analytical sensitivity of <5 pg/mL. The concentration of plasma C-reactive protein was determined using a Cobas Integra 70 analyzer (Roche Diagnostics, Ltd.). The concentration of fibrinogen in plasma was measured using a STA Compact analyzer (Roche Diagnostics GmbH, Mannheim, Germany).

### 2.3. Laboratory Measurements of Coronary Sinus Blood Samples

An additional 10 mL of coronary sinus blood was collected from each included patient to assess the following oxidative/antioxidative balance markers: glutathione reductase (GR), glutathione peroxidase (GPx), glutathione transferase (GST), superoxide dismutase (SOD), its mitochondrial isoenzyme (MnSOD), its cytoplasmic isoenzyme (Cu/ZnSOD), catalase (CAT), malondialdehyde (MDA), hydroperoxide lipids (LPH), and ceruloplasmin (CER).

The activity of GR was determined by Richterich’s kinetic method and is expressed as micromoles of NADPH utilized per minute normalized to one gram of hemoglobin (IU/gHb) [10]. The inter- and intra-assay CVs were 2.1% and 5.8%, respectively. The activity of GPX in erythrocytes was determined by the kinetic method of Paglia and Valentine, with t-butyl peroxide serving as a substrate; the results are expressed as micromoles of NADPH oxidized per minute and were normalized to one gram of hemoglobin (IU/gHb) [11]. The inter- and intra-assay CVs were 3.4% and 7.5%, respectively. The activity of GST was measured by Habig and Jakoby’s kinetic method and is expressed as micromoles of thioether formed per minute and normalized to one gram of hemoglobin (IU/g Hb) [12].

The activity of SOD was determined by the Oyanagui method [13]. In this method, XO catalyzes the oxidation of hypoxanthine to xanthine and generates superoxide in the presence of oxygen. Increasing the activity of SOD in the samples caused a decrease in the superoxide concentration and a reduction in the color of the products (naphthyl ethylenediamine and sulfanilic acid). The activities of the SOD isoenzymes MnSOD and CuZnSOD were measured with the use of potassium cyanide as an inhibitor of the CuZnSOD isoenzyme. CuZnSOD activity was taken as the difference between the total SOD activity and the MnSOD activity. The activity of SOD was calculated against a blank probe (containing bidistilled water). Enzyme activity is expressed as nitrite units (NUs) per milliliter of serum. NU inhibited the formation of nitrite ions by 50% under these conditions. The inter- and intra-assay CVs were 2.8% and 5.4%, respectively.

The activity of CAT was measured by the Aebi kinetic method using a Shimadzu UV-1700 spectrophotometer [14]. It is expressed in IU/mg Hb. The inter- and intra-assay CVs were 2.6% and 6.1%, respectively.

The concentration of MDA was determined with Ohkawa’s method using the reaction of lipid peroxides with thiobarbituric acid and spectrofluorimetric detection. It is expressed in μmol/L [15].

The serum LPH concentration was determined using Södergren’s method. In this method, the oxidation of iron(II) to iron(III) occurs, resulting in a blue–purple complex with xylenol orange [16]. The LPH concentration is expressed in μmol/L.

The concentration of CER was determined according to the spectrophotometric Richterich method [17]. CER catalyzes the oxidation of colorless p-phenylenediamine to a blue–violet dye. The test sample contained twenty microliters of serum, whereas the control sample contained 20 μL of serum; 200 μL of sodium azide solution was added to stop the reaction. In the next step, 1 mL of p-phenylenediamine dihydrochloride in acetate buffer was added to both samples. After a 15 min incubation, 200 μL of sodium azide was added to the test sample. Finally, after 15 min of incubation, the absorbance of the test and control samples was measured at 560 nm using a PerkinElmer VICTOR-X3 plate reader (Waltham, MA, USA). The intra-assay CV was 3.7%, and the intra-assay precision was 4%. The CER concentration is expressed in mg/dL.

### 2.4. Statistical Analysis

The statistical analysis was performed with SAS software (version 9.4). Continuous variables are expressed as the mean ± standard deviation or median with upper and lower quartiles, according to the variable distribution. Continuous variables were compared using the independent Student’s *t* test for normally distributed variables and the Mann−Whitney U test for nonnormally distributed variables. Categorical variables are expressed as numbers (percentages) and were compared using the chi-square test. A *p* value < 0.05 was considered to indicate statistical significance.

Univariable logistic regression analysis was used to select potential factors associated with all-cause mortality for inclusion in the multivariable analysis, with particular emphasis placed on oxidative/antioxidative balance parameters. The examined covariables included CR, LPH, CAT, GPX, SOD, fibrinogen, urea, sodium, erythrocyte sedimentation rate, and creatinine. The relationships between variables were evaluated by Spearman’s rank correlation coefficient. We tested different model variants. The presented models are the result of the best fit of factors to the models, accounting for clinical and statistical significance. The results are presented as odds ratios with 95% confidence intervals.

A receiver operating characteristic (ROC) curves were constructed to determine the utility of the factors associated with one-year mortality obtained from multivariable analysis. The Youden index was used to determine the optimal cutoff point for each factor. The prognostic power of the analyzed parameters was assessed by the area under the curve (AUC), sensitivity, specificity, negative predictive value, positive predictive value, and accuracy.

## 3. Results

Our study cohort included 85 HT candidates (New York Heart Association (NYHA) classes III (80%) and IV (20%); Interagency Registry for Mechanically Assisted Circulatory Support (INTERMACS) score of 4–6). During the one-year follow-up, the mortality rate was 40% (*n* = 34). The basic demographic, clinical, and laboratory data of the patients in the study cohort divided into survival and nonsurvival groups are presented in Table 1.

Multivariable stepwise logistic regression analysis revealed that lower coronary sinus CAT and CER concentrations, as well as higher creatinine concentrations, were independently associated with worse prognosis in patients with advanced HF. The summary of the univariable and multivariable logistic regression analyses is presented in Table 2.

The coronary sinus CER and CAT showed excellent prognostic power, sensitivity, and specificity for identifying patients at increased risk of one-year mortality. In turn, the peripheral creatinine concentration had acceptable sensitivity and specificity for the assessment of prognosis in patients with advanced HF. The ROC curves for CER, CAT, and creatinine are shown in Figure 2A–C.

The results of the ROC analysis for the analyzed factors are shown in Table 3.

## 4. Discussion

Our single-center prospective study showed that lower CER and CAT levels in coronary sinus blood and higher creatinine concentrations in peripheral blood were associated with a greater mortality rate in patients with advanced HF. CER and CAT had excellent predictive power, sensitivity, and specificity for identifying patients at high risk of death in the analyzed population. In turn, the creatinine concentration had acceptable prognostic strength and sensitivity, as well as poor specificity, for the assessment of one-year mortality on the HT waiting list.

This is the first study to demonstrate a strong association between CER concentration in the coronary sinus and the risk of mortality in patients with advanced HF. CER is a multifunctional protein synthesized mainly in the liver that transports approximately 95% of the total circulating copper to cells for a broad range of protective effects, including the activities of cytochrome c, superoxide dismutase, and metalothioneins [18,19]. CER is characterized by an ambivalent nature and has both antioxidant and pro-oxidant properties, which depend on the environmental conditions, substrate changes and differential localization or expression [19,20]. The antioxidant property of CER results from its activity as an oxidase, which is related to the possible oxidation of ferrous ions to less toxic ferric forms; the control of biogenic amine levels in intestinal fluids and plasma; and the scavenging of ROS such as singlet, superoxide, and hydroxyl radicals [19,21]. In turn, the oxidative properties of CER may be related to its action as a nitric oxide (NO) oxidase, resulting in decreased NO levels [22]. CER may play an important regulatory role in the heart and in the development and progression of cardiovascular diseases. Interestingly, the uptake of CER by the myocardium was demonstrated in an animal model, and the total and specific binding of CER was 2–7 times greater in the heart and brain than in the liver. This finding suggested that CER is an important enzyme in the heart [23]. Mateeseu et al. showed that CER can protect cardiac tissue against the damaging effects of free oxygen radicals [24]. Similarly, Chahine et al. reported that in isolated rat hearts, CER has a protective effect against electrolysis-induced ROS [25]. Another study also showed that CER can also protect endothelial cells against the harmful effects of ROS and regulate the degree of cell and protein damage caused by oxygen radicals [20]. Under these conditions, an increase in CER reserves may contribute to an increase in the copper level, and a decrease in CER will be detrimental when copper reserves are deficient [26,27]. In turn, copper deficiency can lead to systolic and diastolic heart dysfunction in association with typical histopathological changes (increased collagen accumulation in copper-deficient hearts along with lipid deposition), which are indices commonly used to diagnose congestive HF [28].

The exact mechanisms of the cardioprotective effects of CER are not fully understood but are likely related to its oxidase activity, including extracellular antioxidative ferroxidase, glutathione (GSH)-peroxidase, and NO oxidase/S-nitrosating activities, which allow for the scavenging of ROS [29,30]. CER is an important plasma ferroxidase that catalyzes the oxidation of ferrous ions to less toxic iron species (conversion of Fe^2+^ to Fe^3+^) and inhibits the Fenton reaction, which uses Fe^2+^ to produce oxygen radicals [19]. In turn, CER reduction may trigger the release of cellular iron and contribute to oxidative damage caused by ROS [31]. CER also has other important biological functions, including inhibiting lipid oxidation and preventing damage to proteins and genetic material [18,19]. CER can also mimic cytoplasmic superoxide dismutase and catalyze the dismutation of superoxide anion radical (O_2_^•−^) to oxygen and H_2_O_2_, which is further eliminated by catalase or glutathione peroxidase [32]. When the CER concentration is reduced, the process of removing toxic superoxide anion radicals is impaired. Superoxide anions can be released in various biological processes, including autoxidation reactions or by enzymes involved in aerobic metabolism (the leakage of electrons from the electron transport chain in the mitochondria). However, the production of superoxide anions and other ROS may significantly increase under various pathological conditions, such as hypoxia or inflammation, two important pathological processes in HF, which promote damage to electron transfer within the mitochondrial respiratory chain [32,33].

Another important finding of our study was the independent association between lower CAT activity in coronary sinus blood and worse one-year mortality in the analyzed population. CAT is the primary enzyme responsible for the detoxification of H_2_O_2_ into oxygen and water. A decrease in CAT activity affects the accumulation of H_2_O_2_. In turn, a high concentration of H_2_O_2_ reduces the dismutation ability of enzymes such as Cu/Zn superoxide dismutase and Fe superoxide dismutase, leading to the accumulation of O_2_^•−^, which inhibits CAT activity [34,35]. The above results show a cooperative interaction between catalase and superoxide dismutase activities. Thus, low concentrations of enzymes that act as superoxide dismutases, such as SOD and CER, result in low concentrations of CAT. Moreover, an imbalance in SOD to CAT activities causes the accumulation of H_2_O_2_, which participates in the Fenton reaction, resulting in the formation of toxic hydroxyl radicals (OH•) [35]. Ide et al. reported a significant relationship between increased production of ·OH, ·O^2−^ and H_2_O_2_ and left ventricular contractile dysfunction [36]. In turn, high catalase activity may protect the heart against adverse remodeling. Qin et al. showed that myocardial CAT overexpression prevents the cellular hallmarks of adverse remodeling of myocytes and the progression to overt HF in Gαq-overexpressing transgenic mice [37]. Conversely, a decrease in CAT activity may lead to the accumulation of H_2_O_2_, which contributes to the dysregulation of multiple redox pathways and may lead to adverse cardiac remodeling by mediating myocyte fibrosis [38,39]. Bäumer et al. reported a significant decrease in CAT activity in explanted hearts from patients with end-stage HF [40]. Another study also demonstrated significant decreases in SOD and CAT activity, as well as increases in GPx activity and ROS and reactive nitrogen species concentrations, in patients with HF after cardioverter-defibrillator intervention compared to those in the control group without device intervention [35]. In this study, among the useful indicators of the oxidative–antioxidative system in patients after device intervention, the activity of CAT had the best discriminatory power (AUC 0.8452, 95% CI 0.6844–1.000, *p* = 0.0004), sensitivity (81.3%), and specificity (95.2%) [35].

In our study, we observed lower CER and CAT concentrations in the nonsurvival group than in the survival group. Similarly, the other analyzed antioxidant enzymes (SOD, MnSOD, Cu/ZnSOD, GPX, and GR) were significantly lower in patients who died during the one-year follow-up. Moreover, the levels of oxidative stress markers (MDA and LPH) were significantly greater in the nonsurviving group than in the surviving group. We speculate that this indirectly indicates the exhaustion of the protective effects of both enzymatic and nonenzymatic antioxidants. As a result, oxidative stress increases significantly and causes damage to cellular lipids, proteins, and genetic material, consequently leading to impairment of the proper functioning of the cells.

Another independent factor for one-year mortality in our study group was increased serum creatinine, which is a simple and easy-to-obtain marker of renal insufficiency. Worsening renal function in patients with advanced HF is particularly attributed to hemodynamic factors related to a decrease in cardiac output, which causes worse renal perfusion, an increase in creatinine concentration, and a compensatory increase in tubular sodium retention, ultimately leading to greater cardiac filling pressure [41,42]. Another cause of worse renal function in patients with HF is increased right atrial pressure, suggesting the importance of venous congestion in renal dysfunction [43,44]. Kidney vein congestion contributes to the activation of the renin-angiotensin-aldosterone system and the sympathetic nervous system and induces pathways related to baroreceptors, neural reflexes, and natriuretic peptides. Renal venous congestion also affects endothelial function and the activation of pro-inflammatory, pro-oxidant, and pro-vasoconstrictive pathways [44]. Another important mechanism of worsening renal function in advanced HF is the chronic inflammatory state associated with sympathetic hyperactivity, which leads to cytokine production, increased oxidative stress, fibrosis, and ventricular dysfunction [45,46,47,48]. Inflammation plays a direct role in the development of renal dysfunction, as it mediates most of the deleterious effects of concomitant disorders, including dyslipidemia, diabetes, hypertension, endothelial dysfunction, anemia, bone mineral metabolism perturbations, and malnutrition [48]. Another important factor associated with worse renal function in patients with HF is the need to escalate diuretic doses in response to increasing symptoms and signs of congestion [49]. The use of high-dose loop diuretics leads to greater fluid loss, exacerbates poor glomerular filling, and reduces drug filtration and delivery into the intratubular filtrate, where loop diuretics act [50,51]. These processes may contribute to the progression of renal dysfunction, hypotension and the development of diuretic resistance [49,51]. In turn, diuretic resistance is associated with an increase in angiotensin II and aldosterone activity [51,52] and may contribute to gradual structural changes in the kidneys, including hypertrophy of epithelial cells in the distal tubules, which increases distal sodium reabsorption and limits sodium excretion and diuresis [49,53,54].

Several limitations of our study should be highlighted. This was a single-center study with a small sample size; therefore, the results should be interpreted with caution. However, the group was homogenous, and patients with factors that could affect the oxidative/antioxidative balance, as well as patients after HT and MCS during follow-up, were not included. Furthermore, we analyzed biomarker concentrations only at the time of inclusion in the study. Repeated measurements were not performed, which could have increased the value of the study. Moreover, therapy was modified according to the clinical status during a follow-up, which could have affected the endpoint. Further analysis is necessary to evaluate the role of oxidative stress in this group of patients. Finally, our analysis concerned only the measurement of oxidative/antioxidative balance markers from the coronary sinus. To comprehensively elucidate the role of oxidative/antioxidative balance within coronary circulation, further analysis should also be extended to include markers from the coronary artery. Considering the limitations of this study, multicenter studies with larger samples are needed to further validate the clinical utility of the analyzed indicators.

## 5. Conclusions

In summary, our study showed that lower CAT and CER levels in the coronary sinus, as well as higher peripheral blood creatinine levels, were associated with a greater risk of mortality in patients with advanced HF during a one-year follow-up. CAT and CER levels had excellent predictive power, allowing for the separation of one-year survivors from nonsurvivors. The prognostic strength of the creatinine level was acceptable in the analyzed group of patients. With the broad availability and low cost of tests for oxidative/antioxidative balance measurements, these findings highlight the need to gain further insights into the underlying pathophysiologic process of advanced HF and oxidative stress.

## Figures and Tables

**Figure 1 biomedicines-12-00662-f001:**
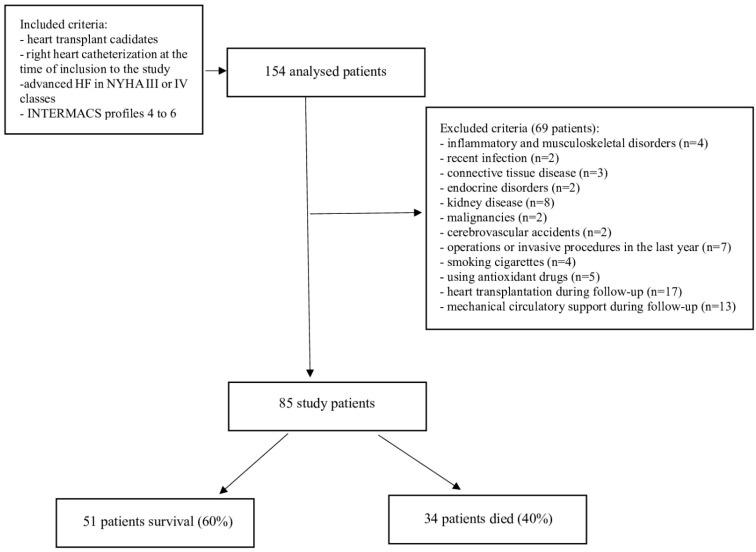
Flow chart of the study design for the inclusion and exclusion criteria. Abbreviations: HF, heart failure; INTERMACS, Interagency Registry for Mechanically Assisted Circulatory Support; NYHA, New York Heart Association.

**Figure 2 biomedicines-12-00662-f002:**
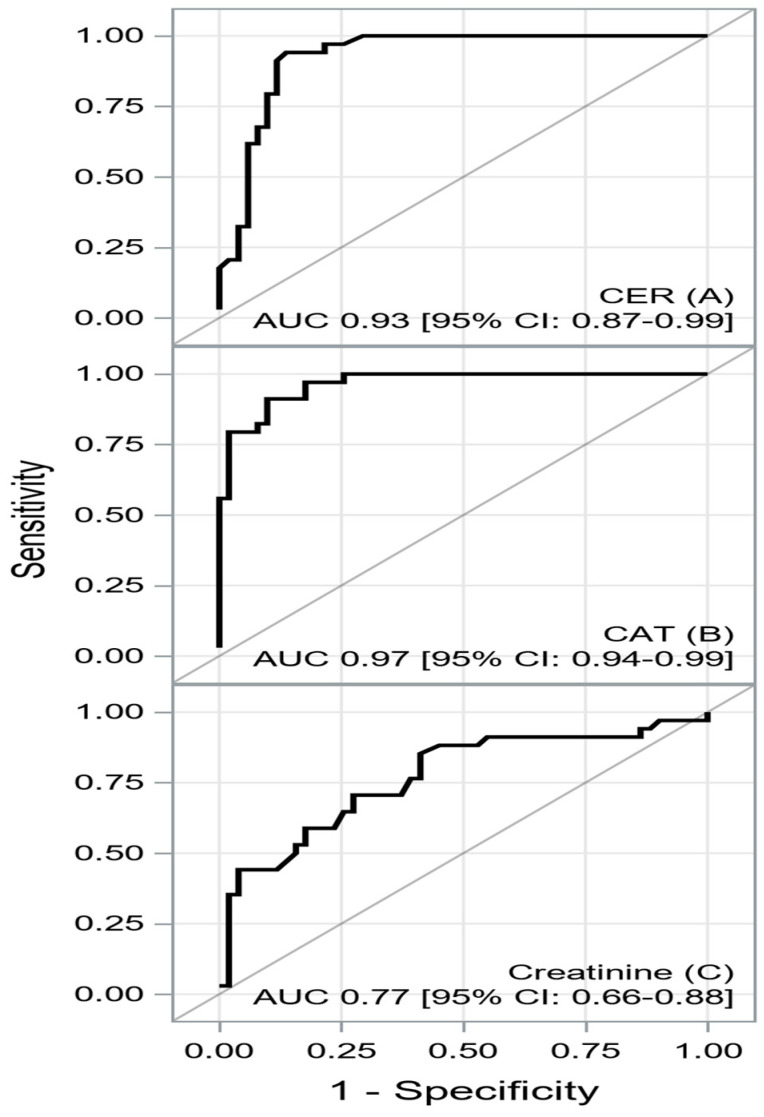
The ROC curves for CER (**A**), CAT (**B**), and creatinine (**C**). Abbreviations: AUC, area under the curve; CI, confidence interval.

**Table 1 biomedicines-12-00662-t001:** Basic characteristics of the study cohort divided into survival and nonsurvival groups.

	Whole Population N = 85 ^#^	Survival N = 51	Nonsurvival N = 34	*p* *
Baseline data				
Age, years	58.00 (50.0–62.0)	58.00 (53.5–61.5)	57.5 (47.2–63.4)	0.7779
Male, *n* (%)	77 (90.6)	45 (88.2)	32 (94.1)	0.3629
Ischemic etiology of HF, *n* (%)	52 (61.2)	34 (66.7)	18 (52.9)	0.2137
BMI, kg/m^2^	27.6 (24.1–30.4)	28.1 (24.1–30.1)	28.9 (25.9–30.1)	0.377
INTERMACS profiles				
Profile 4	15 (17.6)	7 (13.7)	8 (23.5)	0.298
Profile 5	32 (37.6)	18 (35.3)	14 (41.2)	
Profile 6	38 (44.7)	26 (51)	12 (35.3)	
Inotropic support during follow-up, *n* (%)	10 (11.8)	6 (11.8)	4 (11.8)	1.000
Comorbidities				
Hypertension, *n* (%)	50 (58.8)	33 (64.7)	17 (50.0)	0.1771
Type 2 diabetes, *n* (%)	41 (48.2)	22 (43.1)	19 (55.9)	0.2493
Persistent AF, *n* (%)	38 (44.7)	22 (43.1)	16 (47.1)	0.7217
Hypercholesterolemia, *n* (%)	62 (72.9)	38 (74.5)	24 (70.6)	0.6901
Laboratory parameters from peripheral blood samples				
WBC, ×10^9^/L	6.6 (5.4–8.4)	6.7 (5.5–8.3)	6.9 (6.1–7.7)	0.9001
Hemoglobin, mmol/L	8.5 (0.92)	8.5 (0.94)	8.4 (1.2)	0.9311
Creatinine, µmol/L	116.0 (95.0–138.0)	101.0 (90.1–124.2)	134.1 (111.0–159.0)	<0.0001 *
Platelets, ×10^9^/L	182.00 (48.1)	190.1 (48.1)	168.9 (46.75)	0.0603
Total bilirubin, µmol/L	21.4 (14.7–35.5)	20.7 (14.0–35.5)	23.3 (17.00–34.00)	0.5317
Uric acid, µmol/L	477.9 (149.9)	473.8 (145.5)	484.1 (158.3)	0.7702
Urea, µmol/L	9.2 (6.6–13.2)	7.90 (5.4–11.9)	11.7 (8.5–17.1)	0.0068 *
Sodium, mmol/L	139.0 (136.0–141.0)	139.0 (138.0–142.0)	136.0 (135.0–139.0)	0.0003 *
Fibrinogen, mg/dL	382.0 (301.0–482.0)	343.0 (282.0–450.0)	407.0 (375.0–494.0)	0.0005 *
AST, U/L	27.0 (22.2–35.2)	27.0 (23.0–37.0)	27.0 (22.0–34.0)	0.2964
ALT, U/L	25.0 (17.0–36.0)	26.0 (22.00–40.00)	21.1 (16.0–26.0)	0.0129 *
Cholesterol, mmol/L	3.7 (3.4–4.4)	4.2 (3.1–4.7)	3.3 (2.2–4.3)	0.4115
hs-CRP, mg/L	2.7 (1.6–6.1)	1.8 (1.3–4.9)	4.6 (2.7–8.1)	0.0067 *
ESR, mm/h	13.3 (6.8)	9.3 (3.6)	19.4 (5.7)	<0.0001 *
HBA1c, %	6.1 (0.8)	5.8 (0.8)	6.2 (0.8)	0.1651
NT-proBNP, pg/mL	3788.0 (2017.0–5799.0)	3131.0 (1687.0–6473.0)	4307.0 (3352.0–5670.0)	0.0729
Oxidative/antioxidative parameters from coronary sinus samples				
CAT kIU/g Hb	453.4 (363.0–597.3)	58.1 (466.8–684.3)	345.5 (313.7–392.2)	<0.0001 *
SOD NU/mL	20.1 (17.3–23.5)	21.9 (3.9)	19.7 (3.4)	0.0075 *
MnSOD NU/mL	13.2 (10.1–17.7)	15.7 (12.1–20.5)	11.9 (8.7–16.6)	0.0051 *
CuZnSOD, NU/mL	6.2 (4.8–8.2)	7.0 (4.8–8.3)	5.7 (4.7–7.6)	0.2078
LPH μmol/L	1.3 (0.8–1.9)	0.8 (0.6–1.6)	1.9 (1.6–2.4)	<0.0001 *
GR, IU/g Hb	10.2 (9.0–10.8)	10.3 (10.0–11.3)	10.0 (8.5–10.5)	0.0437 *
GPX, IU/g Hb	51.4 (37.8–63.9)	59.9 (20.3)	41.1 (12.7)	<0.0001 *
GST, IU/g Hb	0.36 (0.29–0.42)	0.36 (0.28–0.41)	0.37 (0.30–0.43)	0.4746
CER, mg/dL	40.2 (35.8–47.7)	46.7 (42.8–49.8)	35.6 (32.2–38.0)	<0.0001 *
MDA, μmol/L	3.7 (1.1)	3.0 (0.5)	4.6 (0.84)	<0.0001 *
Hemodynamic parameters				
mPAP, mmHg	27.5 (8.9)	27.6 (9.7)	27.3 (7.9)	0.8565
CI, L/min/m^2^	1.8 (0.2)	1.8 (0.2)	1.8 (0.2)	0.6113
TPG, mmHg	8.0 (7.0–11.0)	8.0 (7.0–10.0)	8.0 (7.0–11.0)	0.8854
PVR, Wood units	2.1 (1.9–2.5)	2.1 (1.9–2.4)	2.2 (1.8–2.6)	0.5435
Spirometry				
FEV1, %	77.9 (13.4)	76.9 (14.6)	79.4 (11.5)	0.4169
FVC, %	83.0 (74.0–92.0)	83.0 (74.0–88.0)	85.5 (75.0–94.0)	0.1875
FEV1/FVC, %	97.0 (93.0–102.0)	98.0 (94.0–103.0)	97.0 (92.0–101.0)	0.6739
Echocardiographic parameters				
LA, mm	54.6 (7.8)	52.8 (8.4)	57.5 (5.9)	0.0032 *
RVEDd, mm	32.00 (30.0–35.0)	32.0 (29.0–35.0)	33.0 (30.0–35.0)	0.3608
LVEDd, mm	75.1 (9.8)	73.2 (9.8)	77.8 (9.3)	0.0326 *
LVEF, %	16.00 (15.0–18.0)	16.0 (15.0–18.0)	16.0 (15.0–18.0)	0.7297
Cardiac medications				
B-blockers, *n* (%)	85 (100)	51 (100)	34 (100)	1.000
Metoprolol succinate dose, mg/day	100.00 (100.00–150.00)	100.00 (100.00–150.00)	100.00 (100.00–150.00)	0.9915
Bisoprolol dose, mg/day	7.50 (5.00–10.00)	7.50 (5.00–10.00)	6.25 (5.00–10.00)	0.5959
Carvedilol dose, mg/day	50.00 (25.00–50.00)	37.50 (25.00–50.00)	50.00 (25.00–50.00)	1.000
ACEI/ARB, *n* (%)	83 (97.6)	50 (98.0)	33 (97.1)	0.7702
Perindopril dose, mg/day	5.00 (5.00–7.50)	5.00 (5.00–7.50)	5.00 (5.00–7.50)	1.000
Ramipril dose, mg/day	5.00 (5.00–7.50)	5.00 (5.00–7.50)	5.00 (5.00–7.50)	0.8499
Valsartan dose, mg/day	160.00 (80.00–320.00)	160.00 (120.00–240.00)	160.00 (80.00–320.00)	1.000
Loop diuretics, *n* (%)	82 (96.5)	48 (94.1)	34 (100)	0.1499
Furosemide dose, mg/day ^^^	120.00 (80.00–160.00)	120.0 (80.0–160.0)	120.0 (80.0–160.0)	0.889
MRA, *n* (%)	85 (100)	51 (100)	34 (100)	1.000
Spironolactone dose, mg/day	50.00 (25.00–50.00)	50.0 (25.0–50.0)	50.0 (25.0–50.0)	0.7672
Epleronone dose, mg/day	50.00 (25.00–50.00)	50.0 (25.0–50.0)	50.0 (25.0–50.0)	0.3497
Digoxin, *n* (%)	33 (38.8)	20 (39.2)	13 (38.2)	0.9276
Digoxin dose, μg/day	0.25 (0.10–0.25)	0.18 (0.10–0.25)	0.25 (0.10–0.25)	0.8494
Ivabradine, *n* (%)	17 (20)	14 (27.5)	3 (8.8)	0.0354 *
Ivabradine dose, mg/day	10.00 (10.00–15.00)	10.00 (10.00–15.00)	10.00 (10.00–15.00)	1.000
Statins, *n* (%)	70 (82.4)	42 (82.4)	28 (82.4)	1.000
Acetylsalicylic acid, *n* (%)	31 (36.5)	19 (37.3)	12 (35.3)	0.8540
Vitamin K antagonists, *n* (%)	52 (61.2)	33 (64.7)	19 (55.9)	0.4135
ICD/CRT-D, *n* (%)	85 (100)	51 (100)	34 (100)	1.000
Other				
VO_2_ max, mL/kg/min	10. 60 (9.70–11.60)	11.20 (10.20–12.00)	10.20 (9.00–11.40)	0.0094 *

^#^ The data are presented as medians (25th–75th percentiles), means (standard deviations), or numbers (percentages) of patients. * *p* < 0.05 (statistically significant). ^^^ for equivalent doses of furosemide (20 mg of torasemide is an equivalent dose of 40 mg of furosemide). Abbreviations: ACEI, angiotensin-converting-enzyme inhibitor; AF, atrial fibrillation; ALT, alanine aminotransferase; ARB, angiotensin II receptor blocker; AST, aspartate aminotransferase; BMI, body mass index; CAT, catalase; CI, cardiac index; CR, ceruloplasmin; CRT-D, cardiac resynchronization therapy-defibrillator; Cu/ZnSOD, cytoplasmic isoenzyme of SOD; FEV1, forced expiratory volume in 1 s; FVC, forced vital capacity; GPx, glutathione peroxidase; GR, glutathione reductase; GST, glutathione transferase; HBA1c, glycated hemoglobin; HF, heart failure; hs-CRP, high-sensitivity C-reactive protein; ICD, implantable cardioverter-defibrillator; INTERMACS, Interagency Registry for Mechanically Assisted Circulatory Support; LA, left atrium; LDL, low-density lipoprotein; LPH, hydroperoxide lipid; LVEDd, left ventricular end-diastolic dimension; LVEF, left ventricular ejection fraction; MDA, malondialdehyde; MnSOD, mitochondrial isoenzyme of SOD; mPAP, mean pulmonary artery pressure; MRA, mineralocorticoid receptor antagonists; NT-proBNP, N-terminal prohormone of brain natriuretic peptide; PVR, pulmonary vascular resistance; RVEDd, right ventricular end-diastolic dimension; SOD, superoxide dismutase; TPG; transpulmonary pressure gradient.

**Table 2 biomedicines-12-00662-t002:** Univariable and multivariable analyses of predictors associated with one-year mortality.

Parameter	Univariable Regression		Multivariable Regression	
	OR (95% CI)	*p*	OR (95% CI)	*p*
CER ^(−)^ *	1.439 (1.230–1.684)	<0.0001	1.342 (1.019–1.770)	0.0363
LPH ^(+)^ *	7.525 (3.011–18.808)	<0.0001		
CAT ^(−)^ *	1.038 (1.019–1.057)	<0.0001	1.053 (1.014–1.093)	0.0076
GPX ^(−)^ *	1.066 (1.031–1.101)	0.0001		
SOD ^(−)^ *	1.178 (1.039–1.336)	0.0104		
Fibrinogen ^(+) #^	1.008(1.003–1.012)	0.0013		
Urea ^(+) #^	1.121 (1.028–1.223)	0.0097		
Sodium ^(−) #^	1.340 (1.142–1.575)	0.0004		
ESR ^(+) #^	1.667 (1.343–2.070)	0.0261		
Creatinine ^(+) #^	1.037 (1.018–1.057)	0.0001	1.071 (1.002–1.144)	0.0422
VO_2_ max ^(−) #^	1.391 (1.071–1.805)	0.0134		

(^+^) per unit increase; (^−^) per unit decrease; abbreviations: Table 1, CI, confidence interval; OR, odds ratio; (*) parameters from coronary sinus samples; (^#^) parameters from peripheral blood samples.

**Table 3 biomedicines-12-00662-t003:** Summary of the ROC curve analysis for biomarkers.

	AUC [±95 CI]	Cutoff	Sensitivity [±95 CI]	Specificity [±95 CI]	PPV [±95 CI]	NPV [±95 CI]	Accuracy
CER	0.9296 [0.8738–0.9855]	≤39.4	0.94 [ 0.80–0.99]	0.86 [0.74–0.94]	0.82 [0.66–0.92]	0.96 [0.85–0.99]	0.89 [0.80–0.95]
CAT	0.9666 [0.9360–0.9971]	≤426.4	0.91 [0.76–0.98]	0.90 [0.79–0.97]	0.86 [0.70–0.95]	0.94 [0.83–0.99]	0.90 [0.82–0.96]
Creatinine	0.7682 [0.6607–0.8756]	≥105	0.85 [0.69–0.95]	0.59 [0.44–0.72]	0.58 [0.43–0.72]	0.86 [0.70–0.95]	0.69 [0.58–0.79]

Abbreviations: Table 1: AUC, area under the curve; CI, confidence interval.

## Data Availability

The data presented in this study are available upon request from the corresponding author. The data are not publicly available due to privacy restrictions related to the rules in our institution.

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
