# Peer review of "Ceruloplasmin, Catalase and Creatinine Concentrations Are Independently Associated with All-Cause Mortality in Patients with Advanced Heart Failure"

_biomedicines, 2024, doi:10.3390/biomedicines12030662_

Round 1

Reviewer 1 Report

Comments and Suggestions for Authors

The authors found that that lower ceruloplasmin and catalase levels in coronary sinus blood and higher creatinine concentrations in peripheral blood were associated with a greater mortality rate in patients with end-stage HF. The benefit of the study is its prospective design, the weakness of one is single-centre obcervation. However, the findings seem to be impressive and I would like to make several comments.

Mayor concerns

1. The authors should give more information of the clinical status of the patients including CRT / DCV, INTERMAX profile, inotropical enhancement, daily dose of diuretics, concomitant medication, and decision-making of the further strategy including heart transplantaion, bridge-to-transplant, others.

2. INTERMAX profile has previously demosnstrated high accurance and predictive value of clinical outcmome among advanced HF patients, so the authors might compare the likelyhood ration, sensitivity, specificity of new models with standard model (INTERMAX).

3. Other circulating biomarkers (troponins, NT-proBNP, adrenomedullin), responce to the treatment, a need to inotropic support , etc. should be included as covariants in the analysis and compare with appropriate methods, for instance AUCs, INR, IDI.

4. Sensitivity analysis should performed and widely interpreted.

5. The sections Discussion and Conclusion seem to be re-written with inclusion in the dispution the issues mentioned above.

Minor concerns.

1. Please, use abbreviations, which have been traditionally used, for instance AF instead of FA for atrial fibrillation. Please, check all abbreviations and correct them when relevant.

2. Flow chart of the study design should be added in the section Methods.

3. Please, re-write subsection Results so than after rough Multivariable stepwise logistic regression adjusted log-regression (to INTERMAX scale) occures.

4. Please, add more information for daily doses of the drugs in the subsection Medication

Author Response

We are grateful to the reviewer for your comments, which helped us improve the manuscript. We have modified our manuscript in accordance with the recommendations of the reviewer. We hope that our publication will now be unambiguous and more understandable to the reader.

Reviewer 1.

The authors found that that lower ceruloplasmin and catalase levels in coronary sinus blood and higher creatinine concentrations in peripheral blood were associated with a greater mortality rate in patients with end-stage HF. The benefit of the study is its prospective design, the weakness of one is single-centre observation. However, the findings seem to be impressive and I would like to make several comments.

1st Comment: 1. The authors should give more information of the clinical status of the patients including CRT / DCV, INTERMACS profile, inotropic enhancement, daily dose of diuretics, concomitant medication, and decision-making of the further strategy including heart transplantaion, bridge-to-transplant, others.

1st Answer: We have added information on the INTERMACS profiles, drug dosages, and other drugs received by patients to Table 1. We excluded patients who underwent HT or mechanical circulatory support (MCS) implantation during the follow-up (n=30) to reduce the bias associated with a nonhomogeneous group of patients (this information can be found in the Materials and Methods subsection). We added the following to the study limitations subsection: “We excluded patients who underwent HT or MCS from our study”. Further analysis is necessary to evaluate the role of oxidative stress in this group of patients.At the time of inclusion in the study, the patients were not receiving inotropic drugs. During the one-year follow-up, 11.8% of patients required inotropic treatment; these data have been added to Table 1.

2nd Comment: 2. INTERMACS profile has previously demosnstrated high accurance and predictive value of clinical outcome among advanced HF patients, so the authors might compare the likelyhood ration, sensitivity, specificity of new models with standard model (INTERMACS). Other circulating biomarkers (troponins, NT-proBNP, adrenomedullin), responce to the treatment, a need to inotropic support , etc. should be included as covariants in the analysis and compare with appropriate methods, for instance AUCs, INR, IDI.  Sensitivity analysis should performed and widely interpreted.

2nd  Answer:  In our study, we analyzed only the baseline parameters of peripheral blood and oxidative stress parameters of the coronary sinus. Unfortunately, we did not analyze other markers, such as troponins and adrenomedullin. Among the parameters mentioned by the reviewer, the NT-proBNP level was analyzed.

As requested, we added NT-proBNP and INTERMACS to the multivariable model, and the ORs were 1.000 (0.943-1.061) for a 100-unit increase in NT-proBNP (p = 0.996), 0.478 (0.001-186.5) for INTERMAX 5 vs. 4 (p = 0.81) and 0.004 (<0.001-30.47) for INTERMACS 6 vs. 4 (p = 0.22). We also compared the discrimination capability of the multivariable models, including NT-proBNP and INTERMACS, with that of the original multivariable model.

AUC

Mulitvariable model vs. Mulivariable model + NT-proBNP

0.9942 vs. 0.9442 (no difference), p = 1 (DeLong’s test)

Mulitvariable model vs Multivariable model + INTERMACS

0.9942 vs. 0.9977, p = 0.28 (DeLong’s test)

NRI and IDI (Pencina et al.)

We used the following cutoff values for the risk of death: <20%, 20-40%, and >40%, which are required for NRI calculations.

NRI (Net Reclassification Index)

Multivariable model vs. Multivariable model + NT-proBNP

NRI (95% CI): 0 (0 – 0),

Multovariable model vs Multivariable model + INTERMACS

NRI (95% CI): 0 (-0.054 – 0.054), p = 1,

IDI (Integrated discrimination improvement)

Multivariable model vs. Multivariable model + NT-proBNP

IDI (95% CI): 0 (-0.0001–0.0001), p = 0.59

Multivariable model vs Multivariable model + INTERMACS

IDI (95% CI): 0.019 (-0.018–0.057), p = 0.31

In summary, there were no statistically significant improvements in the discrimination of the models including NT-proBNP or INTERMACS. It should also be emphasized that the database consisted of 85 cases, and there were 34 events (death); therefore, the number of explanatory variables in the multivariable model was limited.

[Pencina et al.] PENCINA Michael J, D'AGOSTINO SR Ralph B, DEMLER Olga V. Novel metrics for evaluating improvement in discrimination: net reclassification and integrated discrimination improvement for normal variables and nested models. Statistics in medicine, 2012, 31.2: 101-113.

3rd Comment: The sections Discussion and Conclusion seem to be re-written with inclusion in the dispution the issues mentioned above.

3rd Answer: Due to the lack of improvement in the statistical significance of the model after adding NT-proBNP and INTERMACS (in accordance with the results above) and the lack of significant differences for these parameters in the survival vs. nonsurvival groups, we did not rewrite the discussion and conclusions subsections or include the results in the response to the reviewers. If the reviewer is correct, the above results have been added to the manuscript.

4th Comment: Please, use abbreviations, which have been traditionally used, for instance AF instead of FA for atrial fibrillation. Please, check all abbreviations and correct them when relevant.

4th Answer: We checked all abbreviations and corrected any mistakes.

5th Comment:. Flow chart of the study design should be added in the section Methods.

5th Answer: We added a flow chart of the study design to the materials and methods subsection.

6th Comment:.  Please, re-write subsection Results so than after rough Multivariable stepwise logistic regression adjusted log-regression (to INTERMACS scale) occures.

6th Answer: Due to the lack of improvement in the statistical significance of the multivariable model after adding NT-proBNP and INTERMACS (in accordance with the results above), we did not rewrite the results section or include the results in the response to the reviewers. If the reviewer considers it as a correct, the above results have been added to the manuscript.

7th Comment: Please, add more information for daily doses of the drugs in the subsection Medication

7th Answer: We added information on the daily doses of the drugs to Table 1.

Reviewer 2 Report

Comments and Suggestions for Authors

This clinical study aims to determine the association between oxidative/antioxidative balance markers and all-cause mortality in heart transplant candidates, and the authors analyzed the association between routine laboratory parameters and outcomes in the study group. This study found that lower coronary sinus catalase and ceruloplasmin levels, as well as higher peripheral-blood creatinine, were associated with higher risk of death during one-year follow-up of patients with end-stage heart failure in their single-center study. Catalase and ceruloplasmin levels had excellent prognostic power, allowing for the successful prediction of survival versus non-survival outcomes among patients with heart failure. The reviewer considers that the authors well performed the present study, but has major comments as follows:

Major comments:

1.       Lines 74-75: The authors have outlined inclusion and exclusion criteria; however, the inclusion criteria were not explicitly detailed. It is essential for the authors to provide a clear delineation of these criteria. 

2.       Regarding the examination of oxidative/antioxidative balance markers, it appears that measurements were solely obtained from the coronary sinus vein. To comprehensively elucidate the role of oxidative/antioxidative balance within coronary circulation, analysis should also be extended to include markers from the coronary artery. How do the authors address this consideration? 

3.       The laboratory parameters presented in Table 1 appear to be derived from peripheral blood samples. It remains unclear whether the oxidative/antioxidative balance markers were assessed in peripheral blood or the coronary sinus vein. For clarity, the authors should specify the source of the blood samples for these measurements, including those detailed in Table 2. 

Minor comments:

4.       Line 83: The formatting choice to present “mineralocorticoid receptor antagonist” in a larger font and underlined raises questions regarding its significance. Could the authors clarify the rationale behind this typographic emphasis? 

5.       Line 93: Why was “on” written in bold? 

Author Response

We are grateful to the reviewer for your comments, which helped us improve the manuscript. We have modified our manuscript in accordance with the recommendations of the reviewer. We hope that our publication will now be unambiguous and more understandable to the reader.

Reviewer 2

This clinical study aims to determine the association between oxidative/antioxidative balance markers and all-cause mortality in heart transplant candidates, and the authors analyzed the association between routine laboratory parameters and outcomes in the study group. This study found that lower coronary sinus catalase and ceruloplasmin levels, as well as higher peripheral-blood creatinine, were associated with higher risk of death during one-year follow-up of patients with end-stage heart failure in their single-center study. Catalase and ceruloplasmin levels had excellent prognostic power, allowing for the successful prediction of survival versus non-survival outcomes among patients with heart failure. The reviewer considers that the authors well performed the present study, but has major comments as follows:

1st Comment:  Lines 74-75: The authors have outlined inclusion and exclusion criteria; however, the inclusion criteria were not explicitly detailed. It is essential for the authors to provide a clear delineation of these criteria.

1st Answer:  We have corrected the text in the materials and methods section regarding the inclusion and exclusion criteria. In addition, we have added a flow chart of the study design for the inclusion and exclusion criteria in Figure 1.

2nd Comment:  Regarding the examination of oxidative/antioxidative balance markers, it appears that measurements were solely obtained from the coronary sinus vein. To comprehensively elucidate the role of oxidative/antioxidative balance within coronary circulation, analysis should also be extended to include markers from the coronary artery. How do the authors address this consideration?

2nd Answer:  We only analyzed the parameters of oxidative-antioxidative balance only from the coronary sinus, which is an important limitation of this study. In the future, we will also consider assessing biomarkers from coronary circulation. In the limitations section of the study, we added the following phrase:Moreover, our analysis concerned only the measurement of oxidative-antioxidant balance markers in the coronary sinus. To comprehensively elucidate the role of oxidative/antioxidative balance within coronary circulation, further analysis should also be extended to include markers from the coronary artery.

3rd  Comment: The laboratory parameters presented in Table 1 appear to be derived from peripheral blood samples. It remains unclear whether the oxidative/antioxidative balance markers were assessed in peripheral blood or the coronary sinus vein. For clarity, the authors should specify the source of the blood samples for these measurements, including those detailed in Table 2.

3rd  Answer: We have modified Tables 1 and 2 as recommended.

4th Comment: Line 83: The formatting choice to present “mineralocorticoid receptor antagonist” in a larger font and underlined raises questions regarding its significance. Could the authors clarify the rationale behind this typographic emphasis?

4th Answer: This was a mistake during the formatting process of the  manuscript according to the journal’s rules.  We have modified our manuscript and chosen the correct font throughout the entire manuscript.

5th Comment:  Line 93: Why was “on” written in bold?

5th Answer: This was a mistake during the formatting process of the manuscript. We have removed the bold font in this line.

Round 2

Reviewer 1 Report

Comments and Suggestions for Authors

The authors submitted a critically revised version of the paper along with thoroughly prepared reply to the reviewers. I have no serious concerns about the paper in its revised version.

Reviewer 2 Report

Comments and Suggestions for Authors

This reviewer has no further comment.